# Hydraulic Vehicle Damper Controlled by Piezoelectric Valve

**DOI:** 10.3390/s23042007

**Published:** 2023-02-10

**Authors:** Lech Knap, Michał Makowski, Krzysztof Siczek, Przemysław Kubiak, Adam Mrowicki

**Affiliations:** 1Institute of Vehicles and Construction Machinery Engineering, Warsaw University of Technology, Narbutta 84, 02-524 Warsaw, Poland; 2Faculty of Mechanical Engineering, Lodz University of Technology, Stefanowskiego 1/15, 90-537 Łódź, Poland; 3Ecotechnology Team, Lodz University of Technology, Piotrkowska 266, 90-924 Łódź, Poland

**Keywords:** controlled damper, piezoelectric valve, experimental investigation, damper characteristic, reaction time

## Abstract

In this paper, an original construction of a vehicle vibration damper controlled by means of a valve based on piezoelectric actuator is presented and investigated. The presented valve allows us to control dissipation characteristics of the damper faster than in other solutions adjusting the size of the gap through which the oil flows between the chambers of the damper. The article also presents the results of the experimental investigation of the above-mentioned damper showing the possibility of changing the value of the damping force five times in about 10 ms by changing the voltage supplying the piezoelectric actuator. Based on these results, dissipative characteristics were determined which enabled the identification of the parameters of the damper numerical model. The article also presents the results of numerical investigations a vehicle model equipped with the developed dampers. The results showed that the developed damper controlled by the use of the piezoelectric actuator can significantly affect vehicle traffic safety by reducing the variation of vertical forces acting on the wheels. The results obtained are so promising that the authors undertook preparations to conduct road tests of a vehicle equipped with the developed dampers.

## 1. Introduction

Recent years have brought a dynamic development in the design and construction of vehicles, including suspensions, caused by the development of electronics and software. New suspension systems are built in a form of cyber–physical systems, consisting of a physical layer, sensors and a cybernetic layer. The physical layer is comprised of various activators (dampers of variable characteristics, actuators, elastic suspension elements), that through sensors (acceleration, displacement, image), are connected with the cybernetic layer responsible for control. This layer has a controller with predefined algorithms responsible for safety and comfort achieved by engaging appropriate activators.

Such solutions are increasingly used not only in road vehicles and aircraft suspensions but also to protect buildings or structures subjected to impact loads [1,2,3,4,5,6,7].

In the case of modern vehicle suspensions, such systems are used to build both so-called semi-active and active suspensions [8]. Semi-active and active suspensions differ in that in semi-active suspensions the main adjustable element is a controlled damper with variable damping characteristics, while the control algorithms of the electronic system are responsible for selecting the appropriate damping force values. In the case of active suspensions, in addition to controlled dampers, other systems are used, such as changing the vehicle’s suspension clearance (indirectly changing the stiffness of the suspension). Another example would be leveling the vehicle’s body roll or changing the values of the forces realized in an activator that combines the properties of the damper and actuator. However, the operation of the active system results in an increase in the energy required for the operation of these suspensions compared to semi-active suspensions. In semi-active suspensions, less energy is required for the operation of the electronic system responsible for changing the characteristics of the controlled damper. For this reason, semi-active is much more popular due to the trade-off between cost and operating efficiency. 

Semi-active and active vehicle suspension systems primarily use the following dampers: hydraulic dampers equipped with electro-valves (EMD), magnetorheological dampers for linear motion (MRD) and rotary motion (MRB) [9,10,11,12,13], electrorheological dampers (ERD) [14] and hydraulic and pneumatic dampers equipped with piezoelectric valves (PZD) [15,16]. Although solutions of this type have many advantages—such as the range of variation of the damping force—the use of magnetorheological fluid and standard solenoid valves results in relatively slow changes in damper characteristics over the full range, as slow as 80–100 ms [17,18,19]. This speed of damper response means that in order to adequately change the characteristics of a vehicle’s damping system, it is necessary to generate a control signal well in advance. This can mean the need to recognize the unevenness of the road in front of the vehicle, which is neither an easy nor reliable task in all weather conditions, and such a solution is also expensive [20]. Therefore, further research is being carried out to develop vibration dampers with a relatively short response time relative to a change in the value of the control signal.

Designing and building an effective vehicle suspension requires not only suitably fast activators with variable properties, but also fast processing of sensor data and generation of control signals in the electronic system. For this, it is also necessary to develop appropriate control algorithms that realize the assumed optimization criterion, for example, comfort criterion. Driving comfort is defined as a reduction in the intensity of changes in vertical acceleration of a selected point of the vehicle body (e.g., driver or passenger seat points), which is defined in ISO 2631 [21,22,23,24]. On the other hand, increased driving safety is influenced not only by the reduction in horizontal and lateral accelerations [24], but also by the change in the value of the forces exerted by the wheels on the road (so-called dynamic wheel load changes) as well as the condition of the tires [25,26,27,28]. When discussing control algorithms, it should be noted that many dampers using non-Newtonian fluid have inherently nonlinear characteristics; therefore, the selection of an effective control strategy remains a difficult problem to solve and apply [29].

In summary, a key issue for an effectively functioning semi-active or active suspension is the construction of a damper characterized by a short response time and with a wide range of damping forces which can be controlled. As mentioned earlier, although there have been many such designs with relatively short response times, there are still investigations undertaken to build dampers that are more effective and with a shorter response time. This is possible due to the new capabilities of sensor and control systems. For example, reducing the response time of the dampers requires identifying road roughness right in front of the car, which is easier than identifying road roughness several or more meters ahead of the vehicle. To meet these requirements, it is necessary, in the case of hydraulic dampers, to develop sufficiently fast controllable valves that allow the damping force to be changed by, for example, throttling the flow through a controlled gap.

For these reasons, in this paper, the authors present the results of undertaken research tasks related to the design, construction and modeling of the characteristics of a controllable damper (PZD-FT) with a piezoelectric valve (PZV). The authors designed the PZV with the use of a piezoelectric actuator. It was necessary to solve the design problems associated with a very short stroke of actuator and changes of stroke in function of force acting on the actuator. It was also necessary to build a fast enough electronic control unit enabling near real-time control of dampers and enabling low-power consumption by harvesting of some of the charge generated on the piezoelectric element when it is loaded. 

The authors used the constructed PZV to build an automotive vibration damper PZD-FT designed for active and semiactive suspensions which next was investigated. The results of performed numerical and experimental studies show that the PZD-FT damper, based on the modification of a Ford Transit van damper, is characterized by a very short response time (8–10 ms) several times shorter in comparison with other solutions based on the use of magnetorheological fluids or proportional valves and offers the possibility of changing the value of the damping force five times as a result of changing the voltage supplying piezoelectric actuator. Thanks to the described properties of the PZDF-FT damper, its concept of design can be used to design and build dampers for other vehicles and others vibration-damping systems. 

In addition, due to the near-to linear characteristics, PZD-FT dampers allow the use of simple control algorithms, as proven by the presented results of numerical simulations. Numerical simulations were carried out using the presented damper model with parameters identified during experimental investigations. The authors also proposed their own algorithm for selecting the value of proper damping forces aimed at increasing the safety of vehicle movement by reducing the variation of the vertical force acting between the road surface and the wheel which can potentially lead to the loss of stability of vehicle movement. Performed numerical simulations show that the developed and discussed control algorithm, although it has its limitations, can be potentially successfully used in vehicle and machine suspension control systems.

## 2. Construction and Operation of PZD-FT Damper

The front suspension of the Ford Transit vehicle is of the McPherson type, and the damper is the guiding element. Therefore, it was decided to design a PZD-FT damper with a PZV valve controlled by a piezoelectric element, which was based on the manufacturer design of the Ford Transit van. Such a solution simplified the process of installing the developed damper in a real vehicle and conducting road tests of the vehicle’s semi-active suspension with operated control algorithms. A general view and a schematic diagram of the PZD-FT damper are shown in Figure 1.

The factory damper belongs to a group of two-pipe vibration dampers with two valves: a bottom valve (1) between the oil chamber and the chamber under the piston and a valve (2) in the piston separating the working spaces above and below the piston. When the damper is compressed, the space under the piston is reduced and it is necessary to pump oil into other working spaces of the damper. The developed damper was filled with hydraulic oil. Oil is pumped through the bottom valve and the piston valve to replenish the amount of oil above the piston (the working space above the piston increases slightly despite the increase in the volume occupied by the piston rod). In this case, the characteristics of the damper are determined by the bottom valve. When the damper is extended, the situation is reversed. Oil from the oil chamber is sucked into the chamber under the piston through the bottom valve, which poses only minimal resistance to the flow of fluid. Oil is compressed in the part of the damper above the piston and its flow through the piston valve determines the damper’s characteristics. The modification of the PZD-FT damper was realized by adding an additional channel with a PZV valve (3), which connects the chamber under the piston to the chamber above the piston. The position of the piston in the valve (3) is controlled by precise control of the piezoelectric element (4).

A general view of the design of the valve with the piezoelectric element (PZV) is shown in Figure 2. Changing the control characteristics of the PZD-FT damper is achieved by controlling the size of the gap (1) in the PZV valve between the housing (2) and the valve piston (3). The space between the valve piston and its housing is sealed by (4). The valve housing has channels (5) and (6) connected to the damper’s internal chambers on both sides of the damper piston. In this way, channels (5) and (6) form a typical bypass channel with respect to the original piston in the factory damper. A piezoelectric element (8) is attached to the pin (7) of the valve piston (3), allowing the size of the gap (1) to be changed quickly and precisely, and this directly alters the damping force realized during damper movement. In order to force the flow through the PZV valve, the flow of oil through the original valves, i.e., located in the piston and the bottom valve, has been limited (Figure 1).

A piezoelectric actuator APA-120L from CEDRAT Technologies [30] was used to control the gap size of the PZV valve, which was connected to the stem (7) of the piston (3). This actuator allows the realization of a 129 μm piston stroke with a supply voltage change from 0 to 150 V and a response time (blocked-free) of about 0.29 ms. Its resonance frequency is equal 1750 Hz in blocked-free mode and 5700 Hz in free-free mode (harmonic excitation). Figure 3 shows a view of the actuator and its characteristics.

The characteristics of the piezo-actuator indicate that the maximum value of the force is realized at the minimum stroke while the realizable value of the force decreases rapidly with the stroke. This is an unfavorable characteristic of the piezo-actuator, because to realize a given force value, it is necessary to limit the value of the piezo-actuator stroke and thus also to limit changes in the size of the gap in the PZV valve. In order to realize a larger stroke value, it is necessary to limit the value of the forces acting on the piezo-actuator (these forces are derived from the pressure of the fluid in the damper chambers). This requirement must be taken into account in the design and construction of the valve and the damper itself, since too high forces acting on the piezo actuator will cause the gap size to decrease or increase on its own. This can lead to obtaining higher or lower damping forces than are desired at the time. 

Therefore, the design and geometric parameters of the valve were chosen so that, due to the formation of hydrodynamic forces from the flowing liquid, the piezoelectric element is not subjected to excessive loading, which could lead to deformation of the piezoelectric element or, in extreme cases, lead to the destruction of the piezo-actuator. Too much mechanical load, compressing the piezo-actuator, results in an electrical charge that can damage the piezoelectric stack. In the design process, it was important to select the diameter of the valve piston and to estimate the value of the forces acting on the piston. For this purpose, a numerical analysis was carried out in Ansys Fluent [31]. Figure 4 shows the results of exemplary numerical simulations of fluid flow through the gap in the PZV valve, which allowed us to estimate the values of pressures and forces acting on the piston and piezo-actuator, as well as selecting the geometric parameters of the gap. The model of the inner fluid in the valve was built with the elements so-called tet10 and wed15 and consisted of about 1250,000 nodes and about 620,000 elements. The results presented in Figure 4 were obtained assuming an initial gap of 0.125 mm, which correspond to a situation in which the piezo-actuator is supplied with the voltage of 150 V (and the valve is closed up to this gap), and the flow rate corresponding to the speed of movement of the piston rod at about 0.03 m/s (which correspond to the inlet velocity of 3.3 m/s and output pressure about 0.2 MPa). It can be seen that the pressure in the valve is distributed fairly evenly although the maximum and minimum values are obtained at the mouth of the inlet and outlet channels. In the situation corresponding to the extension of the damper, pressures reaching a maximum of 3.5 MPa were obtained, while during compression about 0.81 MPa was achieved.

As a result of numerical and experimental studies, a modification of the PZV valve was introduced to allow adjustment of the initial value of the gap. The value of this gap due to the action of the piezo-actuator is reduced, while it is not closed all the way. By adjusting the initial and final value of the gap (without and after the piezo-actuator power is switched on), it is possible to select the minimum and maximum value of the pressure during the movement of the PZD-FT damper and the realized damping force. The flow through such a gap must compensate for changes in the restriction of oil flow through the original valves of the factory damper. For example, if the gap size is too small when the piezo actuator is tuned to its maximum, it may not be possible to pump enough oil into the chambers that increase in volume during damper operation. This can lead to a decrease in pressure in a given chamber, and in an extreme situation can lead to cavitation (evaporation of the fluid at a significant vacuum value), which was confirmed by the results of experimental studies of the characteristics of the PZD-FT damper.

## 3. PZD-FT Damper Properties Testing

Tests of the properties of the factory damper and the designed and built PZD-FT damper with PZV valve were carried out on a dedicated test stand allowing for kinematic excitation—cf. Figure 5. Kinematic excitation of a PZD-FT damper (1) with a built-in PZV valve (2) was realized through a hydraulic actuator (3) controlled by a proportional hydraulic valve with feedback based on the measurement of the displacement of the actuator piston rod. The bench was equipped with sensors: a displacement sensor (4) to measure the displacement of the actuator and indirectly the displacement of the damper, a force sensor (5) and pressure sensors (6) located in the hydraulic channels upstream and downstream of the PZV valve. 

By measuring the displacement of the piston rod and the forces acting on the damper, it was possible to prepare full dissipation characteristics of the PZD-FT damper. The development of these characteristics is necessary to ensure the correct operation of the vehicle suspension system control algorithms during road tests.

Studies of the characteristics of the PZD-FT damper were carried out for different types of excitation, i.e., different amplitudes, velocity of displacement of the piston rod, size of initial gaps in the PZV valve, gaps in the bottom and piston valves, temperatures and voltages controlling the operation of the piezoelectric valve. The frequency of kinematic excitation is ranged between 0.5 Hz and 2.5 Hz. Suspensions are usually designed for frequency between 1 and 1.5 Hz. The experiments on the damper were conducted at the suspension’s resonance frequency and those are the toughest conditions. Forced vibrations of higher frequency will result in lower amplitude of suspension’s vibration. Therefore, the exemplary results were presented for kinematic excitation of 1 Hz, amplitude of 30 mm and are shown in Figure 6, Figure 7 and Figure 8.

Figure 6 shows the course of pressure changes in the channels of the PZV valve. The blue line shows the pressure that is observed in the channel connecting the PZV valve to the chamber above the piston and the red line in the channel leading to the chamber below the piston. It can be seen that the pressure changes sequentially in response to the excitation to which the damper was subjected. During extension, a pressure of about 3.5 MPa is observed while during compression about 0.7 MPa. Thus, the values of the maximal pressure obtained during the experiment are quite similar to those achieved during the numerical simulations shown in Figure 4. Figure 7 and Figure 8 show the damper characteristics achieved at different values of the PZV valve control signal. The characteristics in Figure 7 are shown on the force–displacement of the piston plane and Figure 8 shows the dissipation characteristics on the force–velocity of the piston plane.

Figure 7 and Figure 8 also show a comparison of the characteristics of the factory damper (green) and the PZD-FT damper. For the PZD-FT two extreme characteristics are drawn, respectively, i.e., for no piezo-actuator supply (0 V, blue) and for the piezo-actuator supplied with a voltage of 150 V (red). It can be seen the characteristics of the factory damper are in the extension range of the damper inside the control area of the PZD-FT damper. For the compression region, however, the damping force value is slightly smaller for the factory damper. In addition, the control range of the damping force value for the PZD-FT damper in this area is also limited, due to the need to compensate for changes in the volume of the individual chambers of the damper and the pumping of most of the fluid through the bottom valve (some hydraulic oil is pumped through the PZV valve). The value of the force variation range can be adjusted by the aforementioned initial size of the gap in the PZV valve and by changing the cross-sectional area of the holes in the bottom and piston valves. It can also be noted that the shape of the characteristics of both dampers is similar, and the dissipation characteristics are linear except in the 0–0.15 m/s speed range. In the nonlinear region, the damper is being extended from its compressed position. The explanation for this phenomenon could potentially be twofold. During this operation, oil flows form the oil to chamber under the piston. A slight overpressure is usually applied to the oil chamber by compressing a small amount of gas so that when the oil flows out of the chamber, there is no gauge pressure in the chamber. Overpressure has a higher value when the damper is compressed and decreases as it is extended due to the flow of some oil into the chamber under the piston. The initial value of overpressure causes observed nonlinearity in the characteristics of the damper due to the supported flow of oil from the oil chamber to the chamber under the piston. In addition, these effects can be more pronounced when the flow between the oil and under piston chambers is restricted during the extension of the damper. In such a situation it is possible to observe a gauge pressure in the above piston chamber—in an extreme situation it is even possible to reach the boiling pressure of the oil. It can be observed on the displacement–force characteristics as a vertical step when the movement changes from extension to compression. However, these phenomena can be eliminated by proper selection of gas initial overpressure and a proper selection of valve gaps in the piston. Therefore, in the following part of the article, the linear characteristics of the PZD-FR damper are assumed. This potentially facilitates the selection of damping forces. By changing the supply voltage of the piezoelectric stack of the PZV valve, it is possible to both decrease and increase the damping forces with respect to the damping force realized by the factory damper. 

Based on the obtained dissipation characteristics, an almost fivefold increase in the damping force from the minimum in the PZD-FT damper in the absence of power supply to the maximum when the piezo-actuator is supplied with up to 150 V is evident.

One of the reasons to look for a new damper design is the need to provide the ability to quickly change the damping force value. In the case of the PZD-FT damper, this is possible due to the use of piezoelectric materials and the development of a suitable electronic control system with the ability to accumulate charge when the piezoelectric element is unloaded. Accordingly, part of the experimental research was devoted to the study of the delay time between the change in the value of the control signal and the change in the damping force of the PZD-FT damper. The research was carried out using a real-time system based on the built dedicated electronic circuit for powering piezoelectric elements realizing the fastest possible signal change from the minimum to the maximum value or vice versa. The results of the experimental tests are shown in Figure 9. The analysis of the results shown indicates that the delay in the change of the damping force in the PZD-FT damper resulted from overdriving the valve by a control signal value of 100% (change of the PZ valve supply voltage 0–150 V and 150–0 V) at an average speed of 50 mm/s is about 8–10 ms. Thanks to the use of charge recovery capabilities, the delay time of the control signal value in relation to the value of the piezo-actuator supply voltage, is less than 1 ms. For comparison, similar studies conducted for another type of damper indicated, at a similar speed of piston rod movement, delays of up to 50–67 ms [27]. The longer delay times were mainly due to the different control principles of magnetorheological damper and the delay times of the action of the winding responsible for generating the magnetic field. The obtained overdrive capability times of the PZD-FT damper indicate that the obtained design solution can be used in dynamic systems and vehicle suspensions requiring short response times.

In order to be able to apply the PZD-FT damper in the suspension of a real vehicle, it is necessary to develop a numerical model of the device in such a way that the physical properties are correctly and possibly in accordance reflected in comparison with the results of experimental tests. The knowledge of the model of the PZD-FT damper makes it possible to apply it to the algorithm for selecting damping forces in the vehicle’s suspension, which, based on measurement signals (e.g., suspension deflection) and the adopted optimization criterion, makes it possible to select the appropriate value of the damping force. Therefore, based on the presented experimental results of the basic properties of the PZV-valve damper, it is necessary to determine the numerical model of the controlled damper. This, in turn, is necessary to develop a numerical model of the entire CPS system (e.g., a vehicle suspension model) and to develop appropriate control algorithms. 

The results of the conducted experimental studies indicate that the properties of the PZD-FT damper are linear and that mainly damping plays a dominant role. The influence of the properties of elastic elements and friction is limited and there is no need to model the properties of the PZD-FT damper in the form of a more complex model, e.g., Herschel–Bulkley [32,33], Casson [34,35], or Bingham [33,36]. 

Table 1 lists the values of parameters describing the dissipation properties of the PZD-FT damper, divided into the extension and compression cases of the damper. Figure 10 shows the model dissipation characteristics of the PZD-FT damper based on the identified parameters of the rheological model.

An analysis of the results summarized in Table 1 shows that the values of the damping coefficient in extension change from 1635 Ns/m to 9000 Ns/m. This gives an almost fivefold change in the damping value of the constructed device. In the case of the operation of the PZD-FT damper in compression, the effect is already much smaller and is due to the previously described way of operating the valves in the damper.

It is evident that the numerical model of the damper adopted is very simple and has the linear characteristic, which can be easily used for implementation in the vehicle suspension control algorithm. Due to the simple structure of the mathematical description, calculations based on the developed numerical model can be implemented very quickly, which reduces the delay created in the activator control loop. This implies that it is possible to use the model and the PZD-FT damper in the controlled suspension of a Ford Transit vehicle during traction testing of the vehicle.

## 4. Numerical Studies of the Vehicle Model Equipped with PZD-FT Dampers

The identified model of the PZD-FT damper with a controlled PZV valve was used to conduct numerical tests of a vehicle model with a semiactive suspension. The vehicle model is shown in Figure 11. The value of the desired damping forces realized by the PZD-FT damper can be determined by an optimization algorithm, for example, with an assumed pressure criterion [37]. This allows us to determine the vector determining the friction forces in the suspension of individual wheels of the vehicle. Then, based on the knowledge of the required frictional force on the basis of the characteristics of the PZD-FT damper shown in Figure 10, it is possible to determine the value of the voltage signal feeding the piezoelectric actuators built into the PZV valves of the dampers of individual wheels.

The simplified vehicle model is characterized by a body with mass m and inertia $J_{y}$ as well as system of forces acting on the body: forces in springs $S_{i}$ and dampers $T_{i}$ (where *i* stands for number of wheels). The vibrations of the vehicle model were forced kinematically by the function $\xi \left(t\right)$ with the simplification of the inability of the wheels to detach from the ground, which is typical for vehicles traveling on paved roads. The displacement of the center of mass of the vehicle body is determined by the spatial variable *z* and the angle $\Phi_{y}$ of body rotation with respect to the transverse axis:(1)$$X={\left[z,\;\Phi_{y}\right]}^{T},$$

Based on the vehicle body equation of motion:(2)$$M\ddot{X}+H\left(S+T\right)=0,$$
where *M = diag*(*m*,*J_y_*), it is possible to determine the suspension deflection $U_{i}$ of individual wheels and deformation speed $V_{i}$. In Equation (2), the force of gravity was omitted because the calculations were carried out from a static equilibrium position. Forces in springs *S* and forces in passive dampers *T* are vectors and have the form:(3)S=KU,
(4)T=CV, 
where $K=diag\left(k_{1},k_{2}\right)$, $C=diag\left(c_{1},c_{2}\right)$ and suspension deflection U and deformation speed V are vectors and have the form:(5)$$U=H^{T}X,$$
(6)$$V=H^{T}\dot{X}.$$

However, the vector *H = [H*_1_*, H*_2_*]* defines the configuration space of the action of forces $S={\left[S_{1},S_{2}\right]}^{T}$ and $T={\left[T_{1},T_{2}\right]}^{T}$, which is defined in the form:(7)$$H_{1}={\left[1,-a_{1}\right]}^{T},H_{2}={\left[1,a_{2}\right]}^{T},\;$$

Value of damping force during the studies was determined based on the assumed algorithm, where vector *T* is defined as:(8)$$T=\gamma \left(H^{T}X,H^{T}\dot{X}\right),\;$$
where γ is an operator describing the algorithm utilized to determine the control signals, which is dependent on the deflection and deformation speed of the suspension.

The assumed coefficient characterizing change in the ratio of the wheels force variations takes form:(9)$$W=\frac{1}{Q_{st}}\sqrt{\sum_{i=1}^{2}{\left(S_{i}+T_{i}\right)}^{2}},\;$$
where:*Q_st_*—static load caused by vehicle mass, load and passengers;*S_i_*—spring force of suspension in *i*-th wheel and is related to suspension deflection caused by the static load *Q_st_*;*T_i_*—dampening force of the damper for *i*-th wheel.

In the algorithm for determining the values of damper supply signals, it is possible to use the optimization of the objective function W, which determines the variation of the sum of the forces applied by individual wheels to the road surface. The optimization problem involves minimizing the value of the argument under the root in relation (9) and can be represented as follows:(10)$$T_{w}=\begin{array}{c} \arg\min\\ T\in \Omega \left(V\right) \end{array}\left\{\sum_{i=1}^{2}{\left(S_{i}+T_{i}\right)}^{2}\right\},$$

Solving this problem makes it possible to determine the values of the damping forces that should be realized by the shock absorbers of each wheel, i.e., $T_{w}=\left[T_{1},T_{2}\right]$. It is possible to obtain a set of solutions due to the fact that the optimization function is strictly convex. Then, based on the knowledge of $T_{w}$ and the characteristics shown in Figure 12, it is possible to select the voltage values of the control signals. The determined values of frictional forces simultaneously lead to the minimization of the value of the W index defined by the relation (9) and which is a safety index defining the variation of the vehicle wheel pressure forces on the road surface in relation to the static pressure.

The set of frictional force solutions has constraints due to the damper characteristics and the current strain rate *V*. The set of permissible solutions is defined by the relation:(11)$$\Omega \left(V\right):=\left\{T\in R^{2}:T_{i}\in w\left(V_{i}\right),\;\;i=1,2\right\},\;$$
where $w\left(V_{i}\right)$ is the control signal for PZD-FT dampers at a given suspension deformation speed (Figure 12). The set of permissible control signal solutions is limited by the values of control signals (α, β) corresponding to the permissible values of the voltage supplying the piezoelectric stack (e.g., 0–150 V). The signals are determined independently in compression (index c) and extension (index e) of the PZD-FT damper. The figure shows two areas of solution for $\;\Omega_{e}\left(V\right)$ and $\Omega_{c}\left(V\right)$ for the extended and compressed dampers. Those areas present damping forces that are possible to achieve. In the case of extended damper, the area is contained by two curves, respectively, $\beta_{e}$ (150 V) and $\alpha_{e}$ (0 V) Then, the damping force for a specific damper displacement velocity V is determined based on an algorithm taken from a set of permissible solutions $w\left(V\right)$. It takes the maximum value $T_{e\_max}$ if the point lies on the curve by $\beta_{e}$ or minimal $T_{e\_min}$ if the point lies on the curve $\alpha_{e}$. If the point is located on the vertical line, denoted as set $w\left(V\right)$ then the force will be in the range of $T_{e\_min}<T_{e}<T_{e\_max}$. Whereas, in the case of damper compression, the area is limited by two curves described as $\beta_{c}$ (0 V) and $\alpha_{c}$ (150 V). In this case, the damping force is equal $T_{c\_max}$ if the point is located on the curve denoted as $\beta_{c}$ or $T_{c\_min}$ if the point lies on the curve $\alpha_{c}$. If the point is on the vertical line, denoted as solution set $w\left(V\right)$ then the force will be in the range of $T_{c\_min}<T_{c}<T_{c\_max}$. Then, based on determined forces $T_{e}$ and $T_{c}$ at a given damper displacement velocity V the value of damper control signal and the value of damping coefficient c is determined, which will be, respectively, in the front suspension $c_{1}$ and rear suspension $c_{2}$. The characteristics shown are based on the experimental results of Figure 10. The characteristics of a classic passive damper are also marked, where the damping coefficient is also dependent on the direction of movement of the piston in the damper (compression $c_{c}$ and extension $c_{e}$). Similarly, as in the case of controlled dampers, the damping coefficients are denoted by respective symbols $c_{1}$ (front suspension) and $c_{2}$ rear suspension. Based on the converted vehicle model and the algorithm for selecting the friction force (damping force) and control signal, numerical simulations were carried out in Matlab/Simulink. The vehicle model was characterized by the following parameters: body mass m=1200 kg, momentum of inertia $J_{y}$ = 46,144 kg/m^2^, spring stiffness $k_{1}=k_{2}$ = 47,500 N/m, distance from the center of gravity of the front and rear axle $a_{1}=a_{2}=1.6$ m. For comparison purposes, tests were also carried out at a constant value of the damping coefficient c = const. where the value of the damping coefficient for a passive damper was assumed, i.e., in compression $c_{e}=1310$ Ns/m and extension $c_{c}=4940$ Ns/m. Based on Table 1, the limiting values of damping coefficients α, β also related to the direction of damper movement were adopted, where $\beta_{e}=9000$ Ns/m, $\alpha_{e}=1625$ Ns/m, $\beta_{c}=2930$ Ns/m, $\alpha_{c}=1765$ Ns/m which are related to the limiting values of voltages supplying the piezoelectric stack with voltage $U_{min}$ i $U_{max}$. The numerical results presented in the following section of the article were carried out with kinematic excitation by a sinusoidal harmonic function with amplitude: a=0.02 m and wavelength L=11.12 m. This corresponds to the excitation at a constant speed of 60 km/h (f=1.5 Hz). Figure 13 shows the kinematic excitation waveforms of individual vehicle wheels.

Figure 14 shows the course of permissible damping forces obtained from the limits and the algorithm for selecting the friction (damping) force vector for wheel *i* = 1 (front wheel), respectively. Based on the presented vehicle model for the assumed kinematic excitation, it is possible to determine the forces in the spring elements of the suspension. The values of these forces are shown in Figure 15. In turn, based on these forces and on the basis of the discussed control algorithm, the desired values of damping forces can be determined. The waveform of the damping force values is shown in Figure 16.

To evaluate the performance of the PZD-FT damper and the proposed control algorithm, the course of changes in wheel pressures over time on the road surface was determined based on relation (9). The W index represents the variations of the wheel vertical forces (forces in the suspension) in relation to the static pressure. The higher the value of the index, the higher the dynamics of changes in wheel vertical forces, which affects safety and can lead, at excessive values, to wheel slippage and loss of stability of vehicle movement. Figure 17 shows the course of the W index for the suspension model equipped with PZD-FT dampers and with the original passive dampers. The results show standard deviation of 0.0358 with the semiactive suspension and 0.0617 for the passive suspension. On this basis, it can be concluded that in the case of using a system with variable damping forces and using PZD-FT damper in the analyzed case, an improvement (reduction) of 42% in value of variations of the wheel vertical forces can be obtained. This can significantly affect the traffic safety of the analyzed vehicle.

## 5. Discussion

The experimental and numerical results presented above show that the PZD-FT damper has many advantages, and it may be beneficial to apply it to the suspension of a real vehicle. Although the proposed solution seems to have many advantages, the conducted tests showed that the solution is not without disadvantages. Among the most important are the costs associated with the use of piezoelectric actuators as well as the properties of the actuator itself. The relatively small actuator stroke value is further diminished when the actuator is loaded with forces from hydrodynamic forces. This, in effect, limits the possibility of obtaining greater variability in the minimum and maximum value of the damping force. It is, therefore, necessary to take this problem into account already at the design stage.

Another important problem associated with the use of the PZD-FT damper in vehicle suspension is the susceptibility of the piezoelectric material to atmospheric agents and potential breakdown when moisture is present. This can lead to damage to the actuators. Moreover, sufficiently fast control of the properties of the PZD-FT damper requires the construction of relatively complex power supplies that allow not only rapid change of the control signal but allow protection of the actuator against overload and breakdown. This requires the ability to accumulate charge from the piezoelectric element but this can also be used for energy recovery. Another feature that may hinder the application of the proposed type of PZD-FT damper in vehicle suspensions is the non-typical control voltage (e.g., 150 V) and the power needed to quickly overdrive the PZD-FT damper. For a pair of dampers during experimental testing, the need to provide the power of about 35 W (70 W for 4 PZD-FT dampers) was observed. Rapid overdriving of the piezoelectric element associated with a decrease in voltage can be assisted by projecting a low voltage of the opposite sign. These solutions complicate the power supply system and cause further increase costs.

As it was mentioned, both the valve response time and the linear characteristics make it easier to apply the developed solution in vehicle and machine suspension systems. The linear characteristics will greatly facilitate the application of the damper model in the control algorithm and can significantly reduce the execution time of the algorithm code. The article discusses only one control algorithm which is related to ride safety. The results showed that the use of dampers controlled by the developed algorithm is very beneficial. It should be mentioned that another popular group of control algorithms can be used which is related to providing comfort to the driver or passengers. As a rule, they are based on minimizing the acceleration of the vehicle body at an arbitrarily selected point. As part of further work, it is planned to conduct research on the application of these algorithms not only in the course of numerical simulations but also on a vehicle equipped with PZD-FT dampers.

Nevertheless, based on the many advantages of this solution and the promising results of numerical tests indicating great potential for changing the vehicle’s ride safety, the authors decided to continue research on the developed PZD-FT damper constructions with PZV valves. In the nearest future, the authors plan to conduct road tests which will be aimed at verifying the presented promising results of numerical simulations in a real environment (Figure 18) using different algorithms for selecting damping force.

Figure 18 shows images of the suspension of a van vehicle equipped with the presented PZD-FT dampers (right image). Additionally, the original factory suspension is shown (left image). A comparison of the images shows that the PZD-FT damper is slightly larger because of the relatively large PZV valve built on the outside. Nevertheless, the external difference in design and dimensions is so small that the installation of the PZD-FT damper is not hampered and does not cause any kinematic restrictions, e.g., reducing steering angle.

It is expected that the effect of the planned road tests of the vehicle with controllable PZD-FT dampers will not only bring an increase in driving safety, but also the possibility of extending the working time in the car due to an increase in comfort.

## 6. Conclusions

The article presents the results related to the design, building, and investigation of the controlled damper PZD-FT based on the use of the developed valve with piezoelectric element PZV. The presented valve allows us to efficiently control the dissipation characteristics of the built PZD-FT vehicle damper due to the use of the piezoelectric actuator controlling the gap in the PZV.

Experimental investigation reveals that the developed PZD-FT damper enables unique properties. Due to the use of PZV it is possible to rapidly change properties of the damper in less than 10 ms and to change the value of the damping forces five times at the same time.

Experimental investigations were carried out to determine the characteristics of the PZD-FT damper and identify the values of the parameters of the numerical model of the damper. Based on the vehicle model with the suspension equipped with the identified PZD-FT damper model, simulation studies were carried out. The results showed that with the application of the proposed control algorithm for PZD-FT dampers, it is potentially possible to increase vehicle ride safety, which is illustrated by a change in the proposed load index by about 42%.

The results obtained are promising and the authors undertook preparations to conduct road tests of a vehicle equipped with the developed PZD-FT dampers.

## Figures and Tables

**Figure 1 sensors-23-02007-f001:**
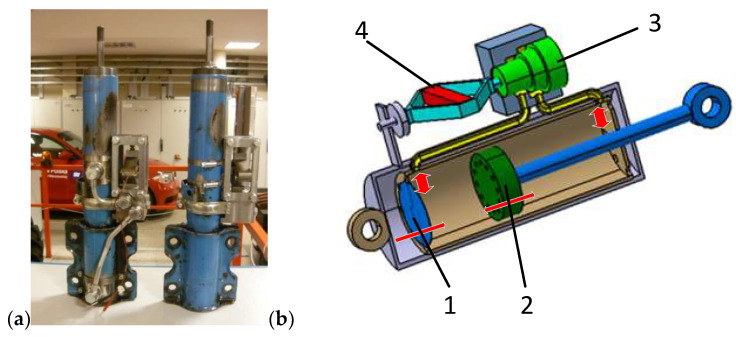
PZD-FT damper: (**a**) PZD-FT damper based on OEM damper design, (**b**) PZD-FT damper diagram: 1—bottom valve, 2—piston with valves, 3—PZV valve, 4—piezoelectric actuator.

**Figure 2 sensors-23-02007-f002:**
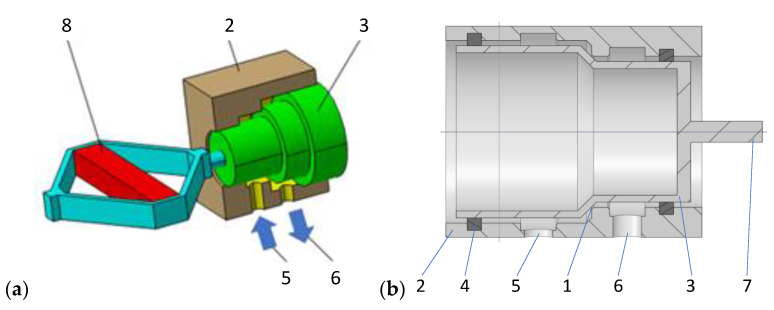
PZV valve: (**a**) valve diagram, (**b**) valve section view: 1—the gap in PZV, 2—housing, 3—piston, 4—sealing, 5—above piston chamber channel, 6—under piston chamber channel, 7—pin, 8—piezoelectric actuator.

**Figure 3 sensors-23-02007-f003:**
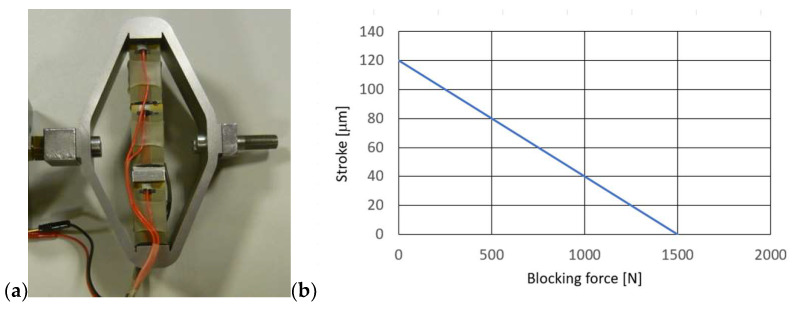
Piezoelectric actuator APA-120L [30], (**a**) view of the piezoelectric actuator, (**b**) characteristic of the piezoelectric actuator.

**Figure 4 sensors-23-02007-f004:**
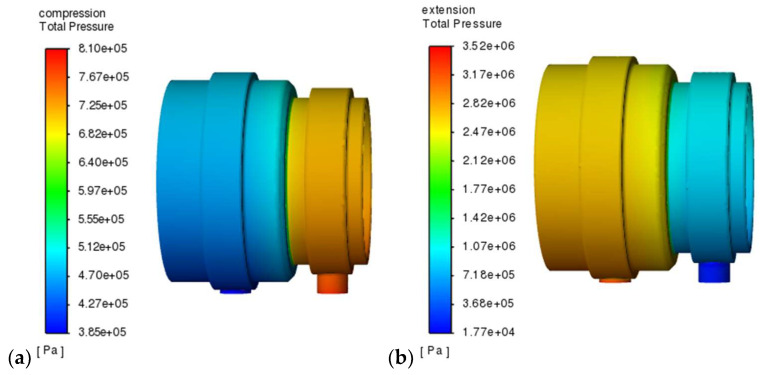
Numerical simulation results of oil flow through the PZV valve: (**a**) damper compression, (**b**) damper extension.

**Figure 5 sensors-23-02007-f005:**
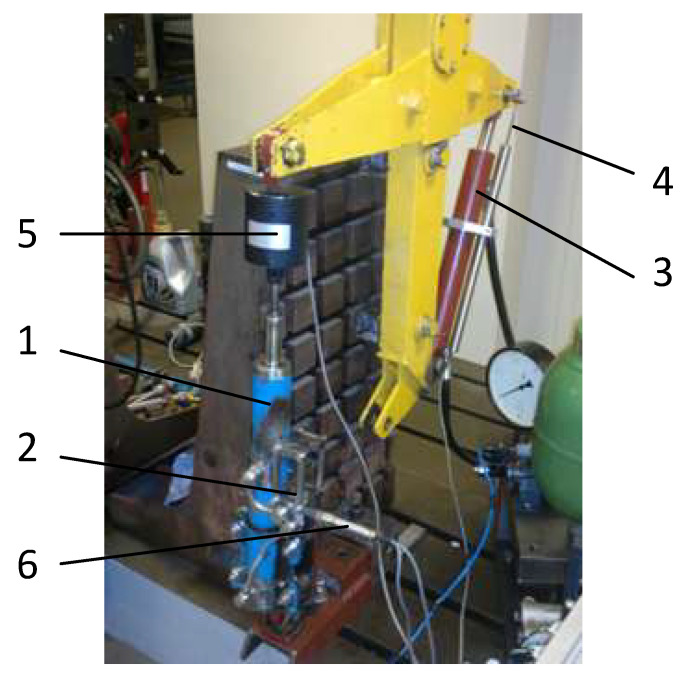
PZD-FT damper built on a test bench: 1—PZD-FT damper, 2—PZV valve, 3—hydraulic actuator working as a kinematic excitation, 4—displacement sensor, 5—force sensor, 6—pressure sensor allowing for pressure readout in the chamber below and above the piston.

**Figure 6 sensors-23-02007-f006:**
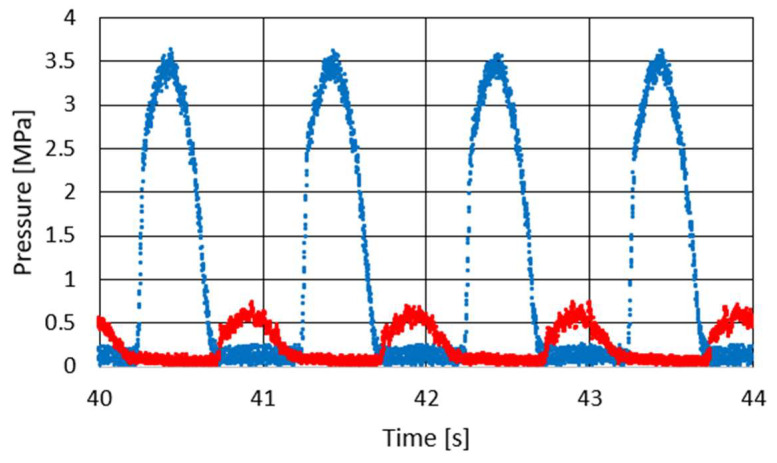
Results of pressure changes on PZV valve channels during experimental testing of a PZD damper with kinematic excitation at a frequency of 1 Hz, amplitude of 0.030 m, voltage of 150 V. The red curve represents the changes of the pressure in channel linked with the chamber below the piston while the blue line represents the pressure in the channel linked to the chamber above piston.

**Figure 7 sensors-23-02007-f007:**
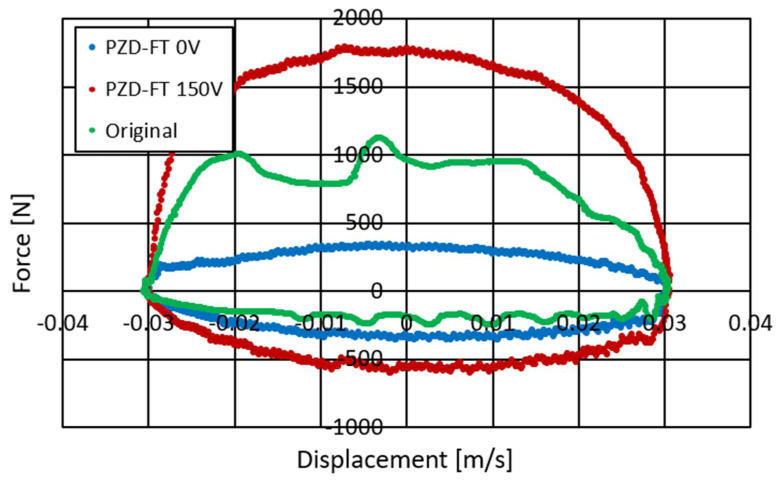
Experimental results of the PZD damper on the displacement-force plane, kinematic excitation with a frequency of 1 Hz, amplitude of 0.030 m, voltage of 0–150 V: PZD-FT 0 V (blue)—PZD without supplying voltage (0 V), PZD-FT 150 V (red)—PZD with supplying voltage (150 V), Original (green) factory passive damper.

**Figure 8 sensors-23-02007-f008:**
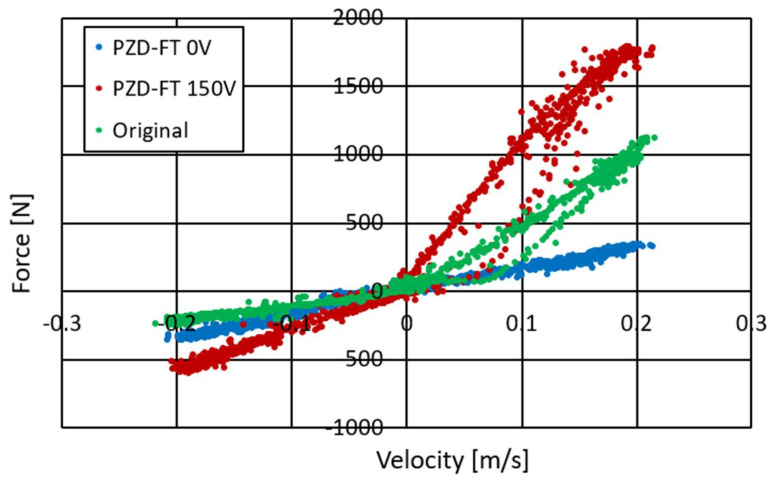
Experimental results of an automotive PZD damper of control in the velocity–force plane, kinematic excitation with a frequency of 1 Hz, amplitude of 0.030 m, voltage of 0–150 V: PZD-FT 0 V (blue)—PZD without supplying voltage (0 V), PZD-FT 150 V (red)—PZD with supplying voltage (150 V), Original (green) factory passive damper.

**Figure 9 sensors-23-02007-f009:**
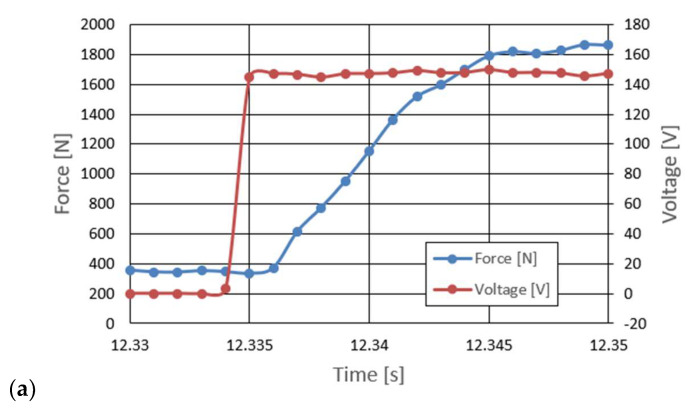
The course of the change of the delay of the damping force and the course of the signal feeding the piezoelectric stack 0–150 V.

**Figure 10 sensors-23-02007-f010:**
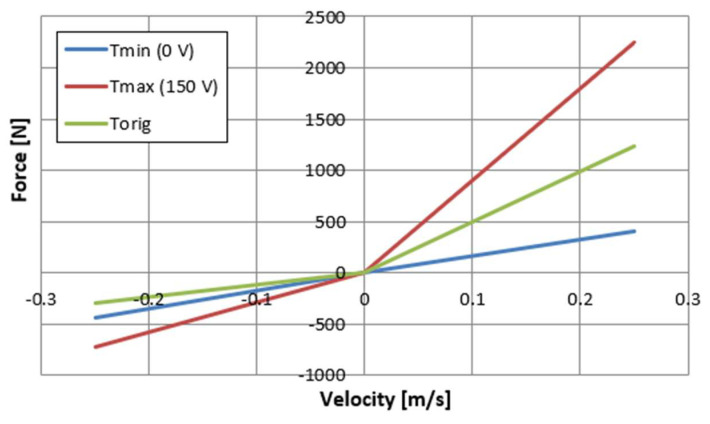
Model characteristics of the PZD-FT shock absorber on the damping force-velocity plane: Tmin (0 V) (blue)—PZD without supplying voltage (0 V), Tmax (150 V) (red)—PZD with supplying voltage (150 V), Torig (green) factory passive damper.

**Figure 11 sensors-23-02007-f011:**
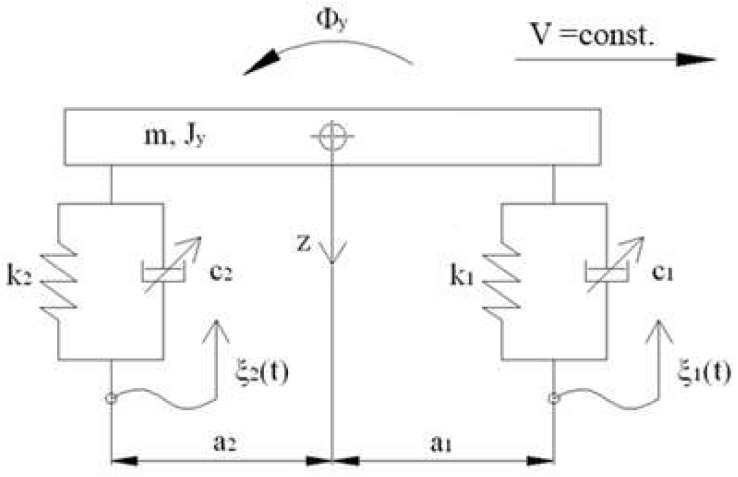
Model of vehicle and semi-active suspension: m—mass of body, J_y_– inertia of body, z—transfer axis, Φ_y_—angle rotation of body, V—velocity of a vehicle, a_1_, a_2_—distance from the center of gravity of the front axle (1), the rear axle (2), k_1_, k_2_—spring stiffness of the front suspension wheel (1), rear suspension wheel (2), c_1_, c_2_—damping coefficient in the front wheel suspension (1), the rear wheel suspension (2), ξ_1_, ξ_2_—kinematic excitation of the front wheel (1), the rear wheel (2).

**Figure 12 sensors-23-02007-f012:**
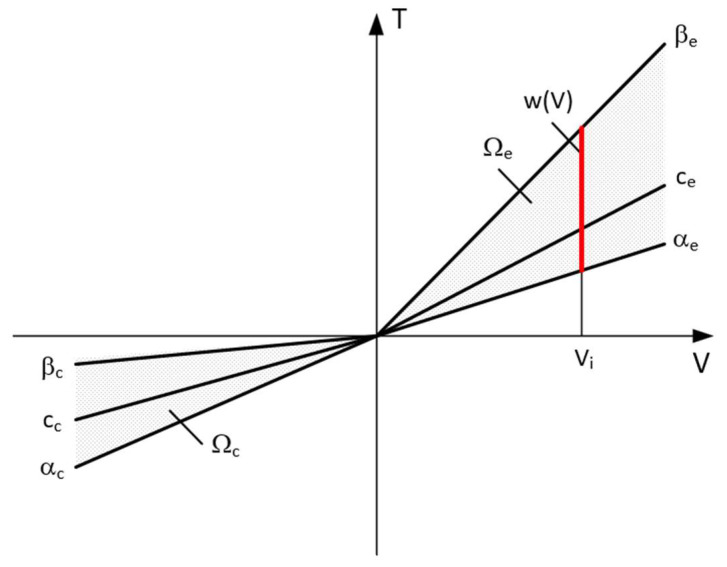
Model characteristics of the PZD-FT shock absorber on the damping force-velocity plane: β_e_—max. extension damping coefficient, β_c_—max. compression damping coefficient, α_e_—min. extension damping coefficient, α_c_—min. compression damping coefficient, c_e_—constant extension damping coefficient, c_c_—constant compression damping coefficient, w(V)—set of accepted control signals.

**Figure 13 sensors-23-02007-f013:**
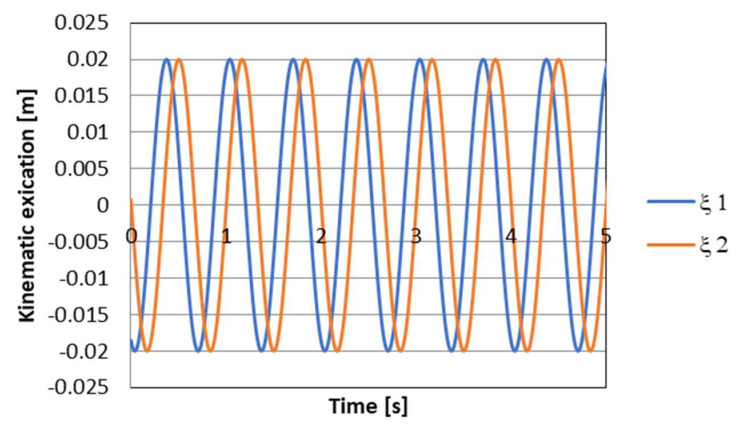
Course of kinematic excitation, respectively $\xi_{1}$—front wheel, $\xi_{2}$—rear wheel.

**Figure 14 sensors-23-02007-f014:**
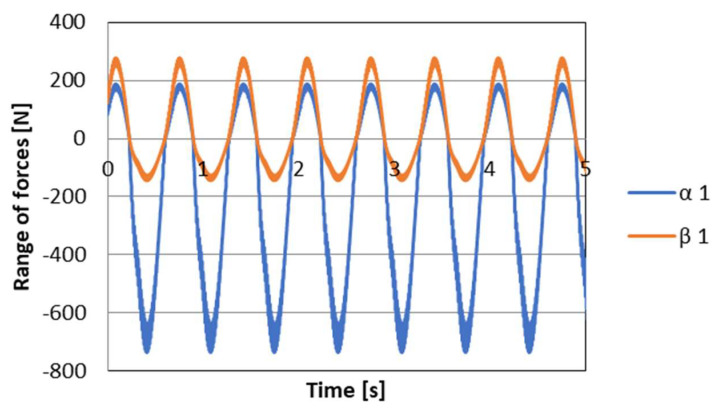
The course of the limits of acceptable frictional forces for wheel 1 (as compared to Figure 12).

**Figure 15 sensors-23-02007-f015:**
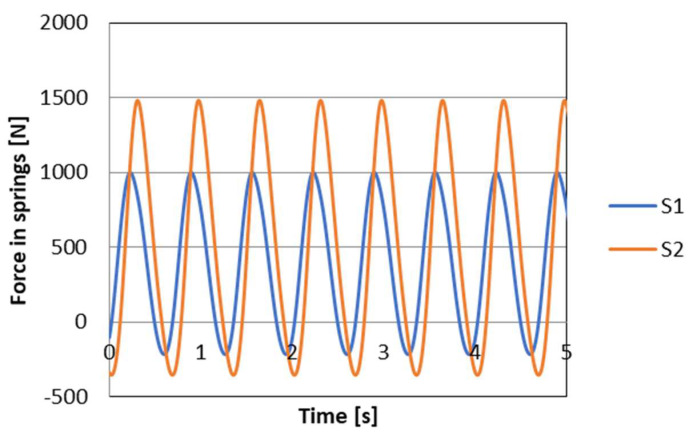
The course of the limit values of forces in the elastic elements of the suspension, respectively S1 front wheel, S2 rear wheel.

**Figure 16 sensors-23-02007-f016:**
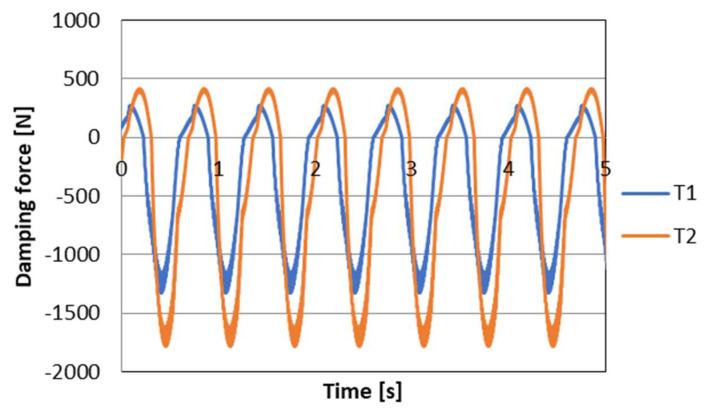
The course of damping forces in the suspension of the vehicle model, respectively, T1 front wheel, T2 rear wheel.

**Figure 17 sensors-23-02007-f017:**
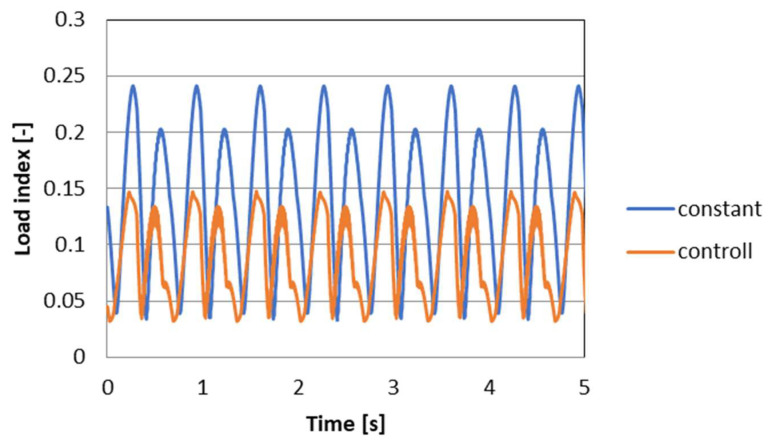
The course of the rate of change of W when controlling the damping force and a constant damping ratio in the suspension of the vehicle model.

**Figure 18 sensors-23-02007-f018:**
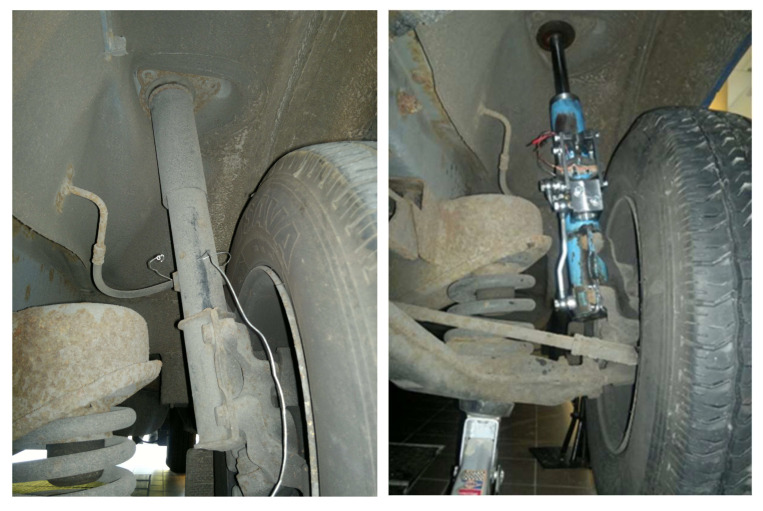
On the left factory damper and on the right PZD-FT damper built into the suspension of a commercial van vehicle.

**Table 1 sensors-23-02007-t001:** Dissipation properties of PZD-FT damper.

Damper	Voltage [V]	Extension (c1) [Ns/m]	Compression (c2) [Ns/m]
PZD-FT	0	1625	1765
PZD-FT	150	9000	2920
Original	-	4940	1310

## Data Availability

The data presented in this study are available on request from the corresponding author.

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
