# Peer review of "Hydraulic Vehicle Damper Controlled by Piezoelectric Valve"

_sensors, 2023, doi:10.3390/s23042007_

Round 1

Reviewer 1 Report

Please provide more description of the experimental results in Figures 7 and 8. Nonlinearity is apparent but not adequately described. Also, the text references orange as factory damper yet this does not appear consistent with the Figure legends.  Please clarify.  This reviewer sees orange, red and blue, but the authors seem to refer to what appears red as orange.  This reviewer suggests using the same curve colors in Figure 10 simulation results as in Figure 8 experimental results for clarity and ease of comparison.  Figure 10 highlights the linearity of the model and the nonlinearity in the Figure 8 experimental results that need to be explained.

Author Response

Thank you very much for your insightful analysis and for pointing out things that might make the article easier to perceive. Indeed, the phenomenon of nonlinearity is important and should be discussed more extensively. During the experimental studies, we identified two factors that could affect the appearance of the nonlinearity area: initial overpressure of gas in the damper and too small gaps in the bottom valve. We have expanded the description of this phenomenon in the discussion section for the results shown in Figures 7 and 8 in lines 238-257. 

 We have also changed Figures 7 and 8 so that the colors match the Figure 10. We have also corrected the description below the figures.

Reviewer 2 Report

The paper presents a hydraulic damper equipped with a valve controlled by a piezoelectric actuator. Simulations and experiments were conducted. Some places need to be revised and explained.

1. The abstract part should highlight your important works with less introductions. L15-26

2. The format of reference number should be revised. For example, it should be [1-7]. L41....

3. Please introduce the structural parameters. L103

4. Please introduce more mesh information of Fig.4. L170

5. Please enlarge the Fig.8 to display the results more clearly. L223

6. Fig.19 is not referenced in the passages. L457

Author Response

Thank you very much for the insightful analysis and for pointing out things that might make the article easier to perceive.  Regarding your comments, the following changes have been made in change tracking mode:

Ad 1. Abstract was changed so it better describes the problem presented in the article.

Ad. 2. We went through the article and corrected the references.

Ad. 3. Description of the structural elements was added for Figures 1 and 2.

Ad. 4. Before the Figure 4, missing information about fluid model and mesh in Ansys/Fluent was added.

Ad. 5. Figures 7 and 8 were enlarged and the colors for subsequent characteristics were unified in Figures 7, 8 and 10. Additional descriptions were added for the course of force shown in the Figures.

Ad. 6. At the end of the article a reference to Fig. 19 was added. Another picture showing a factory suspension was added as well.

Reviewer 3 Report

1. The innovation of this paper must been clearly stated in Abstract and Introduction.

2. The manuscript has a sufficient list of literature; however, it lacks an explanation of the contribution and gap in the state of the controllable damper (PZD-FT) with a piezo-electric valve (PZV), which must be fully described by this study, for example, in the final description on lines 83 to 93.

3. This manuscript may have application value, but its structure and description must be greatly adjusted.

4. The following representations are suggested throughout the text: [1]-[7] -> [1-7] in line 41, [9]-[12] -> [9-12] in line 58, [13], [14] -> [13-14] in line 59, [15], [16] -> [15-16] in line 60, etc.

5. The reviewer suggests that the number of the elements in the Figures must be mentioned in the caption; for example, the captions of Figures 1 and 2 can refer to Figure 5.

6. Please provide relevant specifications about PZD-FT, piezoelectric element. What is the range of frequencies that the PZV valve can perform? Is the kinematic forcing with a frequency range of 1 Hz appropriate? And is it enough to affect the comfort of the driver?

7. More explanation about Figure 4 is require, such as setting parameters, boundaries, gap adjustment and pressure output, etc.

8. More explanation is necessary in Figures 6, 11 and 12. For example, what the red and blue curves represent in Figure 6 and the relationship between parameters k1, k2, c1, c2, ce, cc, ae, ac, be, and bc.

9. Please double check the format of lines 366 to 369.

10. What values a1 and a2 are in the semi-active suspension in Figure 11? Do these differences lead to the variation in timing versus force in springs and damping force in Figures 15 and 16?

11. Try to use Figure 19 to illustrate the difference and effect before and after installing PZD-FT damper into the suspension of a commercial vehicle.

12. The conclusions should be revised. It is a summary of the manuscript, and the quantitative results must be shown. Therefore, the reviewer suggests that the Discussion and Conclusions should be presented separately, and no figures discussed in the Conclusions, for example, Figure 19.

Author Response

Thank you very much for the insightful analysis and for pointing out things that might make the article easier to perceive.  Regarding your comments, the following changes have been made in change tracking mode:

Ad 1. Abstract was changed so it better describes the problem presented in the article and the results.

Ad 2. The end of the introduction was modified, along with the end of the manuscript so it better describes the contributions and advantages and disadvantages of the presented problem.

Ad 3. The contents of the article, especially the abstract, introduction and discussion sections were rewritten, taking into consideration comments from other points.

Ad 4. References were checked and corrected.

Ad 5. Figures 1 and 2 gained descriptions of elements shown on those figures. We hope that now it is clearer and there are no more doubts.

Ad 6. Details about the APA120L actuator were added based on the technical sheet with a link provided [30]. Simulations, as described under Figure 5, were conducted with different frequency of kinematic forcing ranged between 0.5 Hz and 2.5 Hz. Suspensions are designed mainly for frequency between 1 and 1.5 Hz. Therefore, the exemplary results were presented for 1 Hz. The experiments on the damper were conducted at the suspension’s resonance frequency and those are the toughest conditions. Forced vibrations of higher frequency will result in lower amplitude of suspension’s vibration. The ride comfort is assessed based on ISO 2631 norm. There is further numerical and experimental research planned, connected with ride comfort. The ride comfort will be highly dependent on applied algorithms of controlled dampers.

Ad. 7. An additional description of model and simulation conditions were added, as well as the maximum values of pressure.

Ad 8. For Figure 6, the description was improved and text referencing to it was modified. Symbols for drawings 11 and 12 were added as well. Additionally, the relation between parameters k1, k2, c1, c2, ce, cc, ae, ac, be, and bc was explained (lines 450-470)

Ad 9. Line 366 had an editing error. The sentence was corrected and included into the previous paragraph.

Ad 10. Under the Figure 11 a symbol description was added. Figure 15 and 16 show course of forces in springs and dampers, for front and rear wheel. Displacements visible in Figure 15 and 16 are a consequence of different kinematic excitations of front and rear axle.

Ad 11. Figure 18 - another photograph showing the original damper was added, along with a description showing effects and differences.

Ad 12. The end of the article was modified and the discussion and conclusions sections were separated. The text was also rewritten in those two sections.

Round 2

Reviewer 3 Report

The reviewer checked the manuscript of sensors-2165390-v2 carefully. It is apparent that some contents from the manuscript have been improved according to the reviewer’s suggestions. The reviewer’s 2nd comments are recorded as follows:

1. The innovation of this paper must been clearly stated in Abstract and Introduction.

Reviewer’s 2nd comment: Have been corrected.

2. The manuscript has a sufficient list of literature; however, it lacks an explanation of the contribution and gap in the state of the controllable damper (PZD-FT) with a piezo-electric valve (PZV), which must be fully described by this study, for example, in the final description on lines 83 to 93.

Reviewer’s 2nd comment: Have been corrected.

3. This manuscript may have application value, but its structure and description must be greatly adjusted.

Reviewer’s 2nd comment: No comments.

4. The following representations are suggested throughout the text: [1]-[7] -> [1-7] in line 41, [9]-[12] -> [9-12] in line 58, [13], [14] -> [13-14] in line 59, [15], [16] -> [15-16] in line 60, etc.

Reviewer’s 2nd comment: [13], [14] -> [13,14] in line 59 and [15], [16] -> [15,16] are ok.

5. The reviewer suggests that the number of the elements in the Figures must be mentioned in the caption; for example, the captions of Figures 1 and 2 can refer to Figure 5.

Reviewer’s 2nd comment: Have been corrected for clarity, but please double check lines 250 to 285 again. There are large format conversion errors in the PDF file, e.g. “Figure 7 and Figure 8. Figure 6 shows the course of pressure changes…” on line 252.

6. Please provide relevant specifications about PZD-FT, piezoelectric element. What is the range of frequencies that the PZV valve can perform? Is the kinematic forcing with a frequency range of 1 Hz appropriate? And is it enough to affect the comfort of the driver?

Reviewer’s 2nd comment: The reviewer suggests some explanation mentioned on “author's reply” can be placed in the article for the sake of clarity.

7. More explanation about Figure 4 is require, such as setting parameters, boundaries, gap adjustment and pressure output, etc.

Reviewer’s 2nd comment: Corrected according to review comments, but not ideal.

8. More explanation is necessary in Figures 6, 11 and 12. For example, what the red and blue curves represent in Figure 6 and the relationship between parameters k1, k2, c1, c2, ce, cc, ae, ac, be, and bc.

Reviewer’s 2nd comment: Have been corrected.

9. Please double check the format of lines 366 to 369.

Reviewer’s 2nd comment: Have been corrected.

10. What values a1 and a2 are in the semi-active suspension in Figure 11? Do these differences lead to the variation in timing versus force in springs and damping force in Figures 15 and 16?

Reviewer’s 2nd comment: Have been corrected.

11. Try to use Figure 19 to illustrate the difference and effect before and after installing PZD-FT damper into the suspension of a commercial vehicle.

Reviewer’s 2nd comment: Corrected according to review comments, but not ideal.

12. The conclusions should be revised. It is a summary of the manuscript, and the quantitative results must be shown. Therefore, the reviewer suggests that the Discussion and Conclusions should be presented separately, and no figures discussed in the Conclusions, for example, Figure 19.

Reviewer’s 2nd comment: The Discussion and Conclusion sections have been separated and rewritten.

Author Response

Thank you for accepting the list of changes that we did to the text. We have also included the following ones:

Ad 5. We suspect that the formatting errors stem from numbered objects and references to those objects (i.e. figures) and from “track changes” mode. We managed to fix those errors by accepting some of the changes considering figure references and line breaks. As of now, if we generate a pdf there are no repeating figures, large spaces or line breaks between 250-286.

Ad 6. We have modified the experimental research description and kinematic excitation. Part of the proposed text can be found between lines 266 and 271. Description of comfort algorithm and further research was already included in lines 588-593.

Ad 7. In the previous version of the manuscript, the most important information were included. We added information about the inlet velocity and outlet pressure-lines 221 and 222. We understand that the description can be even more in depth, but modeling the valve and conclusions from this modeling could be a material for a separate article. Moreover, this was only preliminary research relating to the initial choice of diameters.

Ad 11. We added an additional paragraph with the expected outcomes of this solution in planned road research. Changes were made in lines 613-615.